# Stateful Dynamics for Training of Binary Activation Recurrent Neural Networks

## Abstract

The excessive energy and memory consumption of neural networks has inspired a recent interest in quantized neural networks. Due to the discontinuity, training binary neural networks (BNNs) requires modifications or alternatives to standard backpropagation, typically in the form of surrogate gradient descent. Multiple surrogate methods exist for feedforward BNNs; however, their success has been limited when applied to recurrent BNNs, but successful when used in binary-like spiking neural networks (SNNs), which contain intrinsic temporal dynamics. We show that standard binary activation approaches fail to train when applied to layer with explicit recurrent weights, and present a theoretical argument for the necessity of temporal continuity in network behavior. By systematically incorporating mechanisms from SNN models, we find that integrative state enables recurrent binary activation networks to reach similar performance as floating-point approaches, while explicit reset and leakage terms do not affect performance. These results show how spiking units enable the training of binary recurrent neural networks and identify the minimally complex units required to make recurrent binary activations trainable with current surrogate methods.

## 1 Introduction

As large neural network models continue to permeate both data center and edge application spaces, counteracting the increasing energy and memory demands of these networks is critical. Quantized neural networks are one such method, which attempt to balance the increased size of the networks with decreased precision in activation functions and weights (Hubara et al., 2017). Today, several software ecosystems exist to perform quantization-aware training and post-training fine-tuning, enabling the reduction of weights and activations to 4-bit or lower resolution for deployment to specialized hardware for inference (Pappalardo, 2023). The limit of decreased precision is binary neural networks (BNN), which may use binary activation functions (BANN) or binary weights (BWNN) (Hubara et al., 2016), and enable extremely energy efficient inference (Zhu et al., 2024). Binary weights and binary activations provide different advantages over their floating-point equivalents. In digital devices, replacing inter-unit activity transmission from floating-point to binary can drastically reduce the large energy overhead of data transfers to and from memory (Orchard et al., 2021). Both activity and weight quantization reduce the memory requirements of networks. In machine learning accelerators based on analog in-memory computing (Aguirre et al., 2024), quantized weights enable tolerance to intrinsic variability in the stored analog memory states, while BANNs can eliminate the complexity and overhead of multi-bit analog-to-digital converters (Xiao et al., 2023).

**Recurrent Binary Activations**  While heavily quantized activations have been demonstrated in a wide range of neural network topologies and BWNNs have been trained in recurrent networks (Alom et al., 2018), standard BANNs have only been reported for feedforward topologies. However, in many edge computing tasks, where the low power consumption of BNNs is particularly desirable, recurrent layers are critical (recurrent neural networks; RNNs). Examples include audio and video processing applications, such as keyword spotting or object tracking. Other use-cases for RNNs may rely on large networks; in video processing, for example, the network size must grow proportionally with the size of the input videos, and therefore require minimized memory consumption per unit. These varied use cases highlight the need for binary recurrent layers under a variety of constraints. Though binary activations are typically absent from the RNN literature, a notable exception is spiking neural networks (SNNs), which utilize temporal dynamics along with binary-valued activations, and are often used as recurrent layers in neuromorphic computing studies (Bittar

& Garner, 2022; Bellec et al., 2018). These networks incorporate loosely biologically inspired temporal dynamics to keep a pre-activation state within each neuronal unit. The state accumulates over time, and the units communicate via binary 'spikes' only when this state reaches a critical threshold.

**Previous Work Training BANNs** Prior methods of training BANNs utilizing backpropagation fall into three broad categories: surrogate gradient descent, probabilistic surrogates, and progressive sharpening. Surrogate gradient descent, which uses an approximation of binary activations during the backward pass, is the most reported and a variant of this method is used to train SNNs (Neftci et al., 2019; Eshraghian et al., 2023). Probabilistic approaches convert the input to an activation layer to a probability function during activity propagation, and perform gradient descent on the underlying probabilities rather than the stochastic activities themselves (Chung et al., 2017). The final method uses a tuneable activation function which progresses from smooth to discrete over the course of training (Severa et al., 2019). Additional methods for training binary activation networks exist, but these use training methods which are more removed from standard backpropagation, such as random perturbations (Bengio et al., 2013; Ma et al., 2023) or more complex biologically-grounded local learning rules that rely on memory-intensive traces of recent activity (Nicola & Clopath, 2017).

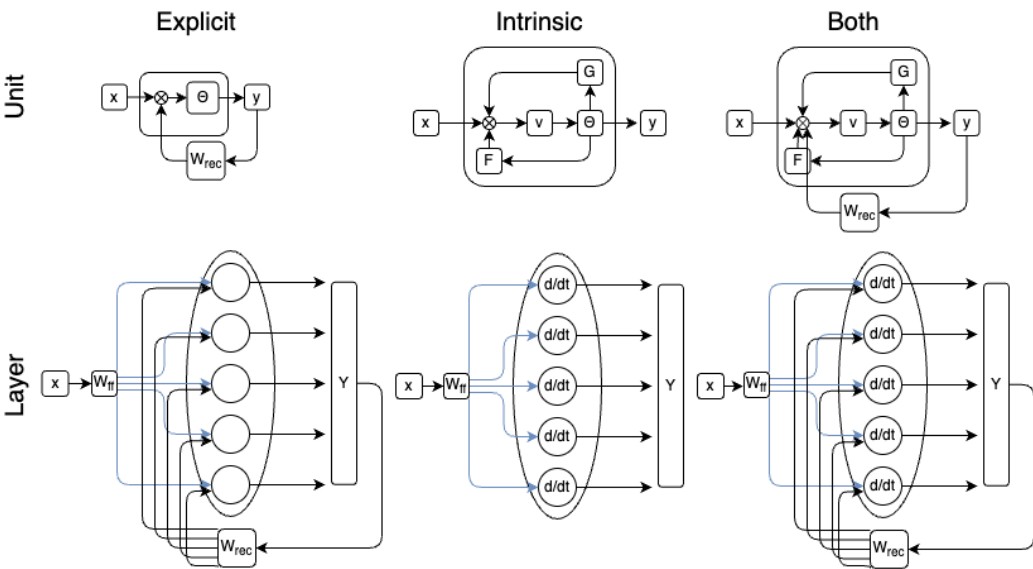

Figure 1: Explicit versus intrinsic recurrence. **Left** In explicit recurrence individual units are feedforward, but recurrent weights allow the layer as a whole to be dynamic. **Middle** Intrinsic recurrent units are each a dynamical system with multiple internal pathways. The layer as a whole however is arranged in a feedforward manner, such that each unit processes inputs independent of the activity of other units. **Right** In combined recurrence dynamical units are arranged in a recurrent topology, resulting in hierarchical levels of temporal processing.

**Explicit Versus intrinsic Recurrence** In order to process temporal relationships among the data, RNNs must have a mechanism for retaining information over time. This mechanism can come in the form of explicit recurrence or intrinsic recurrence, as shown in Fig. 1. **Explicit recurrence**, where the layer's output is used as part of the same layer's input on the next timestep, is more common in standard machine learning frameworks. In the simplest case of a dense recurrent (Elman) layer this follows the form of:

$$y(t + 1) = \Theta(W_{ff}x(t) + W_{rec}y(t)) \tag{1}$$

Where $y$ is the output of of the layer, $\Theta$ is a nonlinear activation function that operates on the sum of weighted inputs, $W_{ff}$ is the feedforward weight matrix that operates on the feedforward activity $x$, and $W_{rec}$ is the recurrent weight matrix. The recurrent matrix takes the output of a neural layer at one timestep and re-presents them on the next timestep, to be summed with input from earlier layer of a network. This ability to mix partially-processed information from previous timesteps with current information, often in a trained manner, allows high-dimensional temporal relationships to be learned. If the recurrent weights of an explicit-RNN were set to zero, the network would operate

as a feedforward network. **Intrinsic recurrence**, in contrast, stores a set of units with time-varying *state*, just as the spiking neural networks outlined above do. An intrinsic recurrent layer may be expressed as the dynamical system:

$$v(t + 1) = F(v(t)) + G(y(t)) + W_{ff}x(t)$$
$$y(t + 1) = \Theta(v(t + 1)) \tag{2}$$

Here the term $v(t)$ represents a *local state* which accumulates information over time without an explicit recurrent connection. $F$ and $G$ are terms which change the state in response to the pre- ($F$) or post-($G$) activation state in the previous timestep. Finally, the nature of this equation prevents the local state ($v(t)$) from instantaneously changing in response to $W_{ff}x(t)$.

Explicit without intrinsic recurrence is typical of standard machine learning approaches such as the gated recurrent unit, where the recurrent weights directly determine the interaction of the previous activity and current inputs. intrinsic without explicit recurrence can act on temporal locality in the processed data, and may occur in applications which emphasize speed of processing, but operate primarily on spatial features with simple temporal patterns (Pedersen et al., 2023; Subramoney et al., 2023; Bing et al., 2020). However, many SNN models utilize both explicit and intrinsic recurrence, which can be written in the general form:

$$v(t + 1) = F(v(t)) + G(y(t)) + H(W_{rec}, y(t)) + K(W_{ff}, x(t))$$
$$y(t + 1) = \Theta(v(t + 1)) \tag{3}$$

This equation includes terms from both equations 1 and 2. In the case of all-to-all layers, the functions $H$ and $K$ are standard matrix multiplication, but may also take other forms, such as convolution. Utilizing this unified equation allows us to drop various terms to more exhaustively investigate the sub-dynamics which are necessary for training.

**Related Work**    As noted above, several lines of research have investigated binary activation functions in feedforward topologies. Additional work has investigated recurrent layers that received binarized inputs, but which still use real-valued activation in recurrent transmission (Edel & Köppe, 2016). Recent work has also combined the explicit recurrence of more advanced networks such as GRUs with intrinsic recurrence, including spike-based transmission, but still use a real-valued pre-activation state (Dampfhoffer et al., 2022). While spiking neural networks have utilized explicit recurrence, the multiple differences between these studies have so far led to an incomplete understanding of the required dynamics for training more general BARNNs.

**Contributions**    Observing that SNNs constitute a subset of BARNNs, specifically combining multiple intrinsic dynamics with explicit recurrent weights, we sought to determine the essential components of SNN-based training which would allow backpropagation-like training. Our main contributions are to:

- Illustrate temporal discontinuities for binary activation explicit recurrent layers, leading to unsuccessful backpropagation through time (section 4.1).
- Demonstrate that surrogate gradient methods fail to converge when employed with a binary activation in a recurrently connected layer (section 4.2).
- Demonstrate, across multiple surrogate approaches, that incorporating pre-activation integrative state allows training of recurrent binary activation networks (section 4.2).
- Show robustness of performance when including additional state dynamics such as explicit reset and proportional leakage of sub-threshold state (section 4.3).

## 2    BINARY ACTIVATION TRAINING METHODS

We implemented three classes of methods which have been used to train binary activation feedforward networks, and compare them amongst each other and with SNN approaches. To enable comparison across these methods, we use the terminology of Equation 3, as summarized in Table 1. Most notably, we will refer to the input to an activation function as $v$. In the case of a standard feedforward network this is simply $Wx$, while for any system with intrinsic recurrence, $v$ is the aforementioned local state. We write the activation function used during the forward pass as $\Theta$ and, where appropriate, the surrogate function used in the backward pass as $\hat{\Theta}$. To further ease comparison, and to allow the same parameter initialization across methods, we utilize activation functions with inflection points centered on zero for each method.

| Variable | Role |
|---|---|
| $x$ | Feedforward input to the layer |
| $W_{ff}$ | Feedforward weight from lower layers |
| $W_{rec}$ | Explicit recurrent weight within the layer |
| $v$ | State of the layer |
| $\Theta$ | Activation function of the layer |
| $\hat{\Theta}$ | Surrogate function of $\Theta$, used for backward pass |
| $\alpha$ | Tuneable parameter that changes over the course of training |
| $y$ | Output of layer, which is forwarded to other layers, and through $W_{rec}$ if present |

Table 1: Standardized variable notations

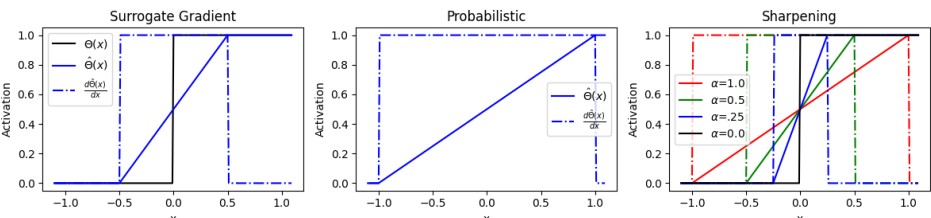

Figure 2: Illustration of the training methods. **Left:** Straight Through Estimate approach, with the true activation function in black, surrogate activation in blue, and derivative of the surrogate as dashed. **Middle:** In the probabilistic training method, the activation function follows a stochastic distribution. The surrogate (solid blue) is a centered and bounded linear function, and the activation on each step is drawn according to a Bernoulli of the function. **Right:** In the progressive sharpening approach, the activation function is a tuneable bounded linear function where the parameter $\alpha$ determines the level of sharpening. At baseline ($\alpha = 1$) this is identical to the standard hardtanh function, and progresses towards the Heaviside function when $\alpha = 0$. There is no surrogate activation for this approach.

**Surrogate Gradient Descent** Of the three approaches, the surrogate gradient descent is the most commonly used and has the most variants, each of which utilizes a different surrogate function ($\hat{\Theta}$) to approximate a non-differentiable activation ($\Theta$) used during the forward pass. Here, we utilized the Heaviside function:

$$\Theta_{ste}(v) = \begin{cases} 1 & \text{if } v \geq 0 \\ 0 & \text{if } v < 0 \end{cases} \tag{4}$$

For the surrogate function, we utilized a variant of the common straight-through estimator (STE) (Hubara et al., 2017):

$$\hat{\Theta}_{ste}(v) = \begin{cases} 0 & \text{if } v \leq -1.0 \\ 0.5 * x + 0.5 & \text{if } |v| < 1.0 \\ 1 & \text{if } v \geq 1.0 \end{cases} \tag{5}$$

which is identical to a shifted hardtanh function and has a piecewise constant derivative.

**Probabilistic** The probabilistic approach takes the pre-activation summed activity $v$ and creates a Bernoulli random variable with mean equal to $v$ (Ma et al., 2023). During the backward pass, the activity is approximated as the real-valued probability ($v$), rather than the stochastic activity. We again utilized a transformation on $v$ before the activation function, in order to achieve zero-centered activity:

$$P(v) = 0.5 + 0.5 * \text{hardtanh}(v)$$
$$\Theta_P(v) \sim \text{Bern}(P(v)) \tag{6}$$
$$\hat{\Theta}_P(v) = P(v)$$

While alternative transformation functions exist, the bounded piecewise linear has the advantage of containing no additional hyperparameters and having a simple piecewise constant derivative with respect to the inputs.

**Sharpening Over Training** In the progressive sharpening approach, no surrogate function is utilized. Instead, the activation function is chosen to be a differentiable and tuneable function, where the tuning parameter causes the function to 'sharpen' and more closely approximate a Heaviside function. The tuning parameter is changed over the course of training such that at the beginning of training the activation function is maximally smooth, and progressively sharpened to the Heaviside at the end of training. Here we will use a tuneable piecewise-linear activation:

$$\Theta_{sharp}(v) = 0.5 + 0.5 * \text{hardtanh}\left(\frac{v + \alpha}{\alpha} - 1\right) \tag{7}$$

Where $\alpha$ is the sharpening parameter. As illustrated in the right-most panel of Figure 2, this is identical to a straight-through estimate within the range $[-\alpha, \alpha]$. When $\alpha$ is 1 this is identical to the shifted standard hardtanh, and when $\alpha$ reaches zero this function is identical to the Heaviside. Following previous approaches (Severa et al., 2019), we start with $\alpha = 1$ and keep it set as such for the first 10 epochs. Afterwards $\alpha$ decreases by 0.01 on each epoch until training error increases by more than 1% greater than the previous minimum, at which point the sharpening is paused for one epoch. When reporting performance of a sharpened network, the metrics are always based on a fully sharpened activation ($\alpha = 0$), regardless of the value at that point in training.

**Spiking Neural Networks** We also train spiking neural networks which implement a binary-like communication scheme in the form of discrete spikes. These networks can be thought of as replacing the feedforward activation functions of standard neural networks with a temporally evolving pre-activation value (state) which integrates the weighted inputs at each point in time. When this state reaches a critical threshold, the unit emits a spike, and resets the state. We specifically utilized the first-order leaky integrate-and-fire (LIF), as implemented via 'LIFBox' in the Norse software package (Pehle & Pedersen, 2021):

$$v(t + 1) = (1 - \frac{g_L}{\tau})(v(t)) - y(t) + \frac{1}{\tau}W_{rec}y(t) + \frac{1}{\tau}W_{ff}x(t)$$
$$y(t + 1) = \Theta_{spk}(v(t + 1)) = v(t + 1) \geq 1 \tag{8}$$

These terms are arranged in the same order as Equation 3. As with the methods above, $v$ is the pre-activation state, but now acts as a stateful leaky-integrator with time-constant $\tau$. $g_L$ is a term which regulates the speed with which $v$ decays to zero in the absence of inputs and is typically referred to as "leak". It is typically set to one, but can also be set to zero to create an non-leaky integrate-and-fire unit. The term $-y(t)$ represents an explicit reset mechanism upon reaching threshold.

The spiking units utilize the "SuperSpike" (Zenke & Ganguli, 2018) surrogate gradient, which operates on the pre-activation state and has the form:

$$\hat{\Theta}_{spk}(v) = \frac{1}{|v + 1|^2} \tag{9}$$

Surrogate gradient methods are then applied as above for each method, where the partial derivative of state ($v$) is now taken through time.

## 3 EXPERIMENTS

We utilize three exemplar tasks, illustrated in Figure 3, to evaluate the performance of the various training methods and demonstrate the required intrinsic recurrence required to train explicit recurrent weights. These tasks are chosen to span static and time-varying domains, with various levels of difficulty, but for which relatively simple recurrent connectivity patterns have shown moderate success. Each task uses a slightly different network architecture, which was chosen for a combination of task appropriateness and simplicity, rather than attempting to utilize state-of-the-art architectures. Data augmentation was not utilized.

**Layers and Hyperparameterization** Each task utilized a different network architecture as outlined below, but all of these are based on convolutional (conv), convolutional-recurrent (crnn), and dense layers. Convolutional layers all use a kernel size of 3, with zero padding to conserve input-output shape consistency. Convolutional recurrent layers contain two convolutional kernel sets, a feedforward and an explicit recurrent (Ballas et al., 2015):

$$H_{crnn}(W_{rec}, y(t)) = W_{rec} * y(t)$$
$$K_{crnn}(W_{ff}, x(t)) = W_{ff} * x(t) \tag{10}$$

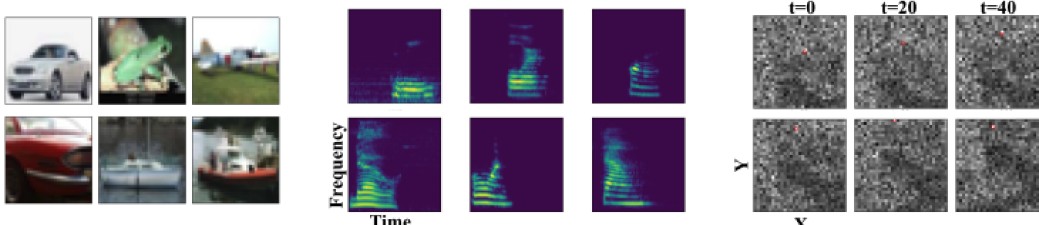

Figure 3: Illustration of the three tasks. **Left:** CIFAR10, a demonstrative static-image classification task. For temporal networks the same image is presented 64 times, and the activity on the final time-step is used to measure performance. **Middle:** Spoken-commands task, demonstrative of temporal classification tasks. The data has been pre-processed to give spectrograms binned into 64 timesteps. Networks are presented with a full spectrum (a column as illustrated here) on each timestep, and the activity on the final time-step is used to measure performance. **Right:** The small-object-tracking task, demonstrative of spatiotemporal tracking tasks. Each image represents the same trial taken at different timesteps. The red circle indicates the position of the target object, for human-reference only. Performance is measured as the L 2 distance between network prediction and ground-truth on each timestep.

Where $*$ is the convolutional operator, and the terms 'H' and 'K' are the explicit recurrence and feedforward activity of Equation 3. Such layers allow recurrent processing in a topologically ordered manner, decreasing the total number of parameters compared to an all-to-all recurrent layer.

The penultimate activation of each network follows a leaky-integrator (LI) equation:

$$v(t + 1) = \left( 1 - \frac{1}{\tau} \right) (v(t)) + \frac{1}{\tau} W_{ff} x(t) \tag{11}$$

Which is similar to Equation 8, but without an additional nonlinear activation. These equations allow integration of inputs from lower layers to form an output based on all time steps without the addition of additional recurrent layers. The time constant $\tau$ was set to four frames in all cases. All other floating-point layers followed the sigmoid activation function, which has the same bounds and inflection points as the surrogate functions described above. For all approaches, training was performed over the course of 200 epochs for five independently initialized random networks, and performance is reported on the average of these. Optimization was performed with the ADAM optimizer (Kingma & Ba, 2017), and learning rates were initialized to 0.001. All other hyperparameters and parameter initializations followed PyTorch defaults (Paszke et al., 2019).

**CIFAR10** (Krizhevsky et al., 2009) The CIFAR10 dataset serves as a baseline to demonstrate the performance of our implementation of binary training methods when applied to non-temporal datasets, as is commonly done in the literature. We utilize a VGG-like architecture (Simonyan & Zisserman, 2014), with [3, 32, 64, 128] two-dimensional convolutional channels, followed by dense layers with [2048, 128, 10] units. An additional CNN-RNN approach replaces the first dense layer with a recurrent layer, allowing comparison to the spiking networks. For networks with temporal dynamics, the entire image was presented for 16 timesteps and the network output was taken as activity on the final step. This repeated input of the image allows activity to travel through the network. CIFAR10 results were optimized with cross-entropy loss and we report top-1 accuracy scores.

**Speech Commands (SC)** (Warden, 2018) This dataset is a temporal classification task, allowing measurement of how well the training methods allow incorporation of spatial convolution with later recurrent processing. Individual trials are clipped, but not explicitly aligned, such that keywords tend to begin about halfway through a trial. Trials which were less than one second long were zero-padded at the beginning of the trial. We preprocess by creating spectrograms with 64 time bins that had 50% overlap and 64 frequency bands from 0 kHz to 4 kHz. On each timestep all 64 frequency bands (a single column of Figure 3B frames) were presented, and the activity of the output layer on the final timestep was used as the network classification. Networks for this task utilized 1-dimensional convolutional-recurrent layers with [1,8,16] channels followed by dense layers with [1024, 128, 35] units, with the exception of the CNN-RNN baseline which utilized 1-dimensional

convolutional layers in place of the convolutional-recurrent layers. These networks were optimized with cross-entropy loss on the output at the final time-step, and we report top-1 accuracy.

**Small Object Tracking (SOT)**   (Chapman et al., 2024) We utilize a recently introduced small object tracking (SOT) task, in which inputs are the frames of a video and target output are the location of a small moving object. This task has been shown to require early recurrent connectivity in neural networks. Briefly, the task utilizes the DIOR remote imaging dataset (Li et al., 2020) and creates a temporal video by introducing camera drift, and pixel-by-time independent noise. This background scene is then scaled [0, 0.5]. A trajectory is then generated which follows a ballistic trajectory with random velocity equal to one fourth of a pixel per timestep. The object's intensity was 0.25 at every time point, giving a single frame a signal-to-noise-ratio of 0.5, and leading to the necessity of early temporal processing. Networks for this task followed a 2-dimensional convolutional(-recurrent) layers with [1,8,16] channels, followed by dense layers with [15376, 128, 2] units, with the final output being the predicted location of the tracked object on a given frame. The loss and reported metric for this task is the mean-square-error (MSE) of readout location averaged across a given 100-frame trial. Importantly, while this loss function is a single value for a given trial, optimization requires minimizing the MSE on each frame, assuring that the task is temporal in nature.

## 4   RESULTS

### 4.1   RECURRENT BINARY ACTIVATIONS ARE DISCONTINUOUS THROUGH TIME

We hypothesized that the recurrent connections of binary activation recurrent networks (BARNN) result in temporal discontinuities which prevent successful backpropagation through time. While surrogate methods accommodate for discontinuities in activation functions, they do no correct for discontinuities in time. To demonstrate this, consider the equation for the gradient of a loss (L) with respect to $W_{rec}$

$$\frac{\partial L}{\partial W_{rec}} = \sum_{t}^{T} \sum_{k=1}^{t+1} \frac{\partial L_{t+1}}{\partial y_{t+1}} \frac{\partial y_{t+1}}{\partial v_{t+1}} \frac{\partial v_{t+1}}{\partial v_k} \frac{\partial v_k}{\partial W_{rec}} \tag{12}$$

This equation notably contains the partial derivative of the output ($y$) with respect to the pre-activation value '$v$', as well as the partial derivative of $v$ with respect to the previous value of $v$, posing two issues. First, in the case of the surrogate gradient training, $\frac{\partial y_{t+1}}{\partial v_{t+1}}$ is implemented as $\frac{\partial \hat{\Theta}(v_{t+1})}{\partial v_{t+1}}$, meaning that this term is an approximation of the activation function. While surrogate gradient training in feedforward networks has been shown to overcome this, when chained through time and multiplied this term will *multiplicatively* increase the error of this approximation. This result is similar to the 'vanishing gradient' issues of recurrent neural networks due to multiplicative terms. Secondly the term $\frac{\partial v_{t+1}}{\partial v_k}$ may be ill-defined if the value of '$v$' changes rapidly. This rapid change in '$v$' can be seen in Figure 4, which compares a trained RNN, BARNN, and SNN. In the case of the RNN and SNN the input and output of the activation function are smooth with respect to time, while the BARNN case shows numerous discontinuities and long periods of static output values. These static values and discontinuities lead to this partial derivative term evaluating to zero on the majority of timesteps.

Table 2 demonstrates how these discontinuous dynamics affect training. The top section of this table reports the performance of a baseline sequential CNN-RNN and a floating-point CRNN (Equation 10) network across all three tasks. The next section reports all three BARNN training methods across our three tasks. All three stateless methodologies show a notable decrease in performance across both temporal tasks (GSC and SOT), while image classification performance is within a few percentage points of the baseline CNN-RNN approach. This confirms previously published results that BANNs, trained using these three methods, can perform non-temporal tasks, but they explicitly demonstrate a lack of convergence for recurrent layers, which has largely been omitted from the literature.

Meanwhile, the SNN-based dynamics of Equation 8 perform substantially better than the stateless BARNN approaches, and only slightly below floating-point CRNN approaches. Previous publications have shown higher performance on the GSC and CIFAR tasks with SNNs, but notably use significantly more complex models with increased number of layers (Bittar & Garner, 2022) or trained delays between units (Hammouamri et al., 2023), which could be combined with the training approaches explored here.

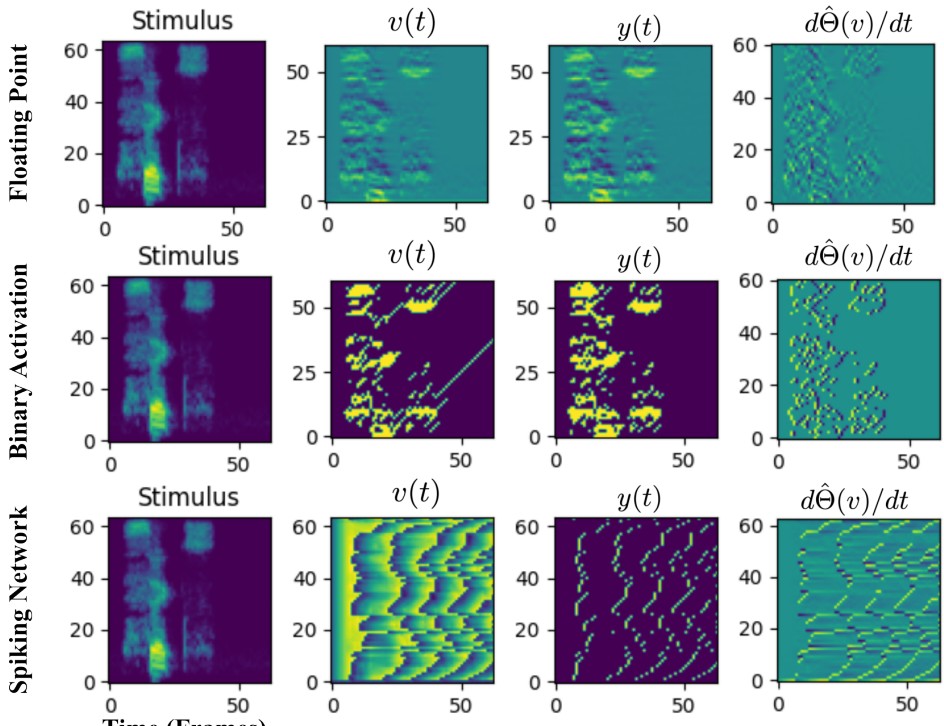

Figure 4: Illustration of activity flow in recurrent networks. Each row indicates activity of a single channel of a convolutional-recurrent network on a keyword spotting task. 'v' is the pre-activation state, which per Equation 3 is simply the summed input activity for the floating point and binary activation rows. 'y' is the corresponding output, and the last column shows the temporal derivatives of surrogate activation functions. **Top:** A standard ('tanh') recurrent network. The pre-activation value 'v' is smooth within each row, indicating a temporally smooth input. Likewise, the output value 'y' is smooth. The final column shows that the surrogate temporal derivatives are less smooth, but take on a number of unique values. **Middle:** Observing the same values for a BARNN-based network, the pre-activation inputs and outputs are largely discrete, due to the recurrent binary activity. The resulting approximate temporal derivatives are discontinuous, with large portions of rows showing no change between timesteps. **Bottom:** In the spiking network, internal dynamics limit the rate of change of 'v' such that it is discontinuous only at spike events. While the output 'y' has discontinuities at event times, the surrogate gradient operation is smooth over time. **Overall:** When performing backpropagation through time the floating-point and spiking network temporal derivatives will be able to differentiable through the majority of timesteps, while the binary activation will primarily be non-differentiable and fail to optimize.

## 4.2 STATE ENABLES RECURRENT ARCHITECTURES

Based on the observations of temporal discontinuities in the previous section, we sought to test whether the stateful dynamics of the LIF activations above can enable training of other recurrent binary-activation layers. For each of the binary training methods (surrogate gradient, probabilistic, and progressive sharpening) the pre-activation input is now stateful and follows the temporal dynamics:

$$v(t+1) = \left(1 - \frac{1}{\tau}\right)v(t) - y(t) + \frac{1}{\tau}W_{rec}y(t) + \frac{1}{\tau}W_{ff}x(t)$$

$$y(t+1) = \Theta_{ste}(v(t+1))$$

(13)

Table 3 reports the results of incorporating these dynamics into the recurrent layers, including the floating-point network to enable fair comparison between networks. We found that the surrogate gradient descent and probabilistic activation functions reach performance on par with the floating-point and LIF-based networks. The progressive sharpening method showed minimal performance increase with the introduction of pre-activation state, and remained approximately as inaccurate as

| | Method | CIFAR10 (Accuracy) | GSC (Accuracy) | SOT (MSE) |
|---|---|---|---|---|
| **FP** | CNN-RNN | 81.44% | 54.96% | 0.091 |
| | CRNN | 84.33% | 81.59% | 0.007 |
| **SNN** | LIF | 81.20% | 78.33% | 0.017 |
| **BARNN** | Sharpening | 82.20% | 49.00% | 0.102 |
| | Surrogate | 78.72% | 29.28% | 0.153 |
| | Probabilistic | 80.23% | 54.96% | 0.095 |

Table 2: Performance of floating point (FP) and binary activations across all three tasks. All networks, with the exception of the sequential CNN-RNN, utilize the CRNN architecture. Metrics reported are averaged over 5 runs and reported on the final values. Performance is substantially impaired in the temporal tasks for BARNNs, but not for LIFs.

the floating-point CNN-RNN method of Table 2. An extensive search was performed on sharpening schedules (see Appendix A), without success. We conclude that pre-activation integrative state is necessary to enable training of binary activations by surrogate gradients and probabilistic activation, and omit sharpening approaches from further analyses below.

| | Method | CIFAR10 (Accuracy) | GSC (Accuracy) | SOT (MSE) |
|---|---|---|---|---|
| **FP** | CRNN-Stateful | 83.24% | 81.86% | 0.006 |
| **SNN** | LIF | 81.20% | 78.33% | 0.017 |
| **Stateful BARNN** | Sharpening* | 72.80% | 67.22% | 0.095 |
| | Surrogate | 79.58% | 80.49% | 0.013 |
| | Probabilistic | 82.30% | 80.00% | 0.014 |

Table 3: Performance of stateful networks across all tasks and training methods. These networks dynamics follow the form of Equation 13 but otherwise have the same architecture as Table 2. Metrics are reported on the final validation values, except for the stateful sharpening approach (marked by $*$) which reports the best performance at any point in training (see Appendix A). Performance of the Stateful BARNN networks, with the exception of the sharpening approach, are close to the FP-stateful entry.

### 4.3 RESETS, LEAKS, AND ACTIVATION VALUES DO NOT CHANGE PERFORMANCE

In addition to an integrative state, standard LIF units of Equation 8 also contain an explicit reset mechanism and a leak term. The SNN literature has shown that a distribution of trainable leak rates can increase the capacity of a network, while explicit resets allow for temporal sparsity of inter-unit communication (Zheng et al., 2024; Zeldenrust et al., 2019). Here we test whether these mechanisms can benefit BARNNs. In the context of the binarization methods introduced here, the reset and leak can be integrated into the state dynamics of Equation 13 to become:

$$v(t+1) = I \times (1 - \frac{L}{\tau})(v(t)) - R \times y(t) + \frac{1}{\tau}W_{rec}y(t) + \frac{1}{\tau}W_{ff}x(t)$$
$$y(t+1) = \Theta(v(t+1))$$
(14)

These dynamics are identical to those in Equation 8, except that the activation function $\Theta$ may now be the STE or probabilistic mechanisms from section 2. The terms 'I','R',and 'L' allow enabling or disabling the integration, reset, and leak mechanisms respectively. Table 4.3 reports the performance across all possible combinations of dynamics-sets. We find no systematic differences in overall performance in any of the tasks.

The dynamics-sets do however show significant differences in learned parameters, as outlined in Appendix B Table 5. One particular learned parameter which we evaluate is the "autapse" which represents the diagonal elements of $W_{rec}$, allowing learned explicit recurrence within a single unit, and which may change pre-activation state on the next step, either strengthen or weaken the effects of intrinsic recurrence. In particular, both the STE and probabilistic approaches contained significantly more non-zero mean weights for the autapses compared to overall distributions of weights. This allows the output activity of a unit to more strongly affect the pre-activation state on the next timestep.

|  | I | L | R | CIFAR10 | GSC | SOT |
|---|---|---|---|---|---|---|
| **LIF** | 1 | 1 | 1 | 81.20% | 78.33% | 0.017 |
| **Surrogate** | 1 | 0 | 0 | 79.58% | 80.49% | 0.013 |
|  | 1 | 1 | 0 | 80.85% | 82.61% | 0.011 |
|  | 1 | 0 | 1 | 80.43% | 77.87% | 0.016 |
|  | 1 | 1 | 1 | 83.61% | 78.23% | 0.017 |
| **Probabilistic** | 1 | 0 | 0 | 82.30% | 80.00% | 0.014 |
|  | 1 | 1 | 0 | 80.22% | 78.89% | 0.012 |
|  | 1 | 0 | 1 | 79.42% | 80.60% | 0.012 |
|  | 1 | 1 | 1 | 80.00% | 76.64% | 0.013 |

Table 4: Performance of binary activation functions with Integration (I), Leak (L), and Reset (R) dynamics. Note that the [1,0,0] (integration-only) configuration is identical to the methods of the previous section.

Overall, STE autapses were significantly positive across all tasks, while probabilistic networks had negative autapse weights. This indicates that STE networks utilize autapses to retain accumulated information, while probabilistic networks tend to self-inhibit after activity. Both of these trends are decreased in dynamics sets containing reset terms, suggesting that the explicit post-activation reset alters the processing of information in each layer.

**Activation Values**  In order to match the {0,1} activation values of typical SNN approaches, our activation functions ($\Theta$) in the above examples have all used {0,1} outputs. However, the standard BNN software packages may default to {-1,1} activation (Pappalardo, 2023) and some SNN work has investigated the use of signed spikes, which are positive or negative events when a local state reaches +/- 1 (Wang et al., 2022), while machine learning has investigated similar three-level activation functions ({-1,0,1}) (Zhu et al., 2024). As a final experiment of robustness of pre-activation state to intrinsic dynamics, we evaluated the stateful-leaky model with these three activations. For the {-1,1} case $\Theta$ was the 'sign' function, and $\hat{\Theta}$ was the same as Equation 5, but recentered about the origin. For the three-level case the surrogate gradient was the same, but the forward activation was:

$$\Theta_{tri} = \begin{cases} sign(v) & \text{if } |v| \geq 0.5 \\ 0 & \text{if } |v| < 0.5 \end{cases} \tag{15}$$

The full table of performance for this experiment is in Appendix C, but overall we found no systematic difference in performance for the GSC task across activation values.

## 5 CONCLUSIONS

We investigated current methods for training binary activation neural networks, and their generalization to recurrent networks. We found that without additional modifications, surrogate-based, probabilistic, and progressive sharpening approaches to train BANNs could not be adapted for recurrent layers due to the temporal discontinuities induced by binary activations. However, incorporating an integrative state smooths these temporal dynamics, and results in successful training in recurrent layers. Additional SNN-based dynamics, such as slow leak and explicit reset, did not substantially alter the performance of such stateful networks. This demonstration, along with the theoretical argument of section 4.1 lead us to conclude that pre-activation state is necessary and sufficient for training of binary recurrent networks.

The ability to train binary activation RNNs has the ability to resolve several constraints currently facing their widespread use. The reduced activation precision can reduce the memory usage of existing CPU/GPU and FPGA accelerators, allowing larger RNN networks to be utilized in both edge and HPC scenarios. These benefits could be further increased by informed quantization of the state (Venkatesh et al., 2024; Apolinario et al., 2024), allowing it to be implemented with a simple register shift. Probabilistic binary activation functions open the possibility of deployment to highly efficient but noisy emerging devices (Ma et al., 2023). When designing a network for inference on a hardware accelerator, the inherent properties of the device will determine the best choice of dynamics which show otherwise show similar performance. These additional findings illustrate how non-standard neural network models can open additional use-cases and influence the design of emerging accelerators.

## 6 REPRODUCIBILITY

We have included the necessary details to ensure the reproducibility of our empirical results. Custom activation functions are described in the equations of section 2. Network topologies, loss functions, and training hyperparameters are described in section 3. Data for the CIFAR10 and Speech Commands datasets are openly available by the cited source papers. Underlying images for the SOT task can be downloaded by the cited DIOR paper, and the methods for generating video data are described in the final paragraph of section 3.

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

## A    APPENDIX: SHARPENING OPTIMIZATION SEARCH

The sharpening paradigm described in section 2 was successful in training feedforward networks on CIFAR10 classification task, but did not perform better than baseline for the recurrent tasks after introduction of pre-activation state. The previous work of Severa et al. (2019) demonstrated that the sharpening approach schedule can have drastic effects on even feedforward networks, leading us to perform an exhaustive grid search of components of the scheduling:

- Sharpening Schedule:
    - Relative error: [0.1%, 1%]
    - Regular Interval (epochs): [1, 2, 5] (maximum epochs adjusted accordingly)
- Initial $\alpha$: [1, 0.5, 0.25]
- $\Delta\alpha$: [.01, .005, .001]
- Optimizers: [ADAM, SGD]
- Learning rate: [1e-3, 5e-4, 1e-4]

In every case we found that training error steadily decreased until approximately $\alpha$ was in the range of [0.2 - 0.15], after which it significantly steadily increased with decreasing $\alpha$ and recovered only slightly with extended periods of frozen sharpening. Validation error (with $\alpha = 0$) was minimal during this period where it matched training error once the training error had significantly increased. It is possible that future work finds additional modifications to the training method, such as population-coded output layers or layer-wise sharpening, that allow the sharpening approach to benefit from pre-activation state for training recurrent layers.

## B    APPENDIX: LEARNED WEIGHTS AND AUTAPSES

|  | Dynamics | | | CIFAR10 | | SC | | SOT | |
|---|---|---|---|---|---|---|---|---|---|
|  | I | L | R | All | Autapses | All | Autapses | All | Autapses |
| Surrogate | 1 | 0 | 0 | 0.00 , .21 | **0.86 , .30** | 0.00 , .11 | **0.13 , .14** | 0.00 , .57 | **3.53 , 1.89** |
|  | 1 | 1 | 0 | 0.02 , .24 | **0.59 , .49** | 0.00 , .10 | **0.11 , .12** | -0.02 , .39 | **2.17 , .91** |
|  | 1 | 0 | 1 | 0.00 , .19 | **0.56 , .51** | -0.00 , .15 | **0.13 , .14** | **-0.07 , .68** | **0.64 , .81** |
|  | 1 | 1 | 1 | **0.09 , .27** | **0.58 , .31** | 0.00 , .08 | 0.01 , .08 | **0.09 , .89** | **1.47 , 3.1** |
| Probabilistic | 1 | 0 | 0 | 0.01 , .20 | 0.08 , .17 | -0.02 , .34 | **-0.37 , .24** | 0.05 , 0.47 | **0.07 , .52** |
|  | 1 | 1 | 0 | 0.01 , .23 | 0.19 , .28 | 0.01 , .19 | **-0.36 , .32** | **-0.11 , .60** | **-0.51 , .55** |
|  | 1 | 0 | 1 | -0.01 , .18 | 0.01 , .28 | 0.05 , .11 | **-0.42 , .48** | **0.04 , .46** | **-0.52 , 0.78** |
|  | 1 | 1 | 1 | 0.00 , .23 | 0.07 , .15 | **0.12 , .34** | **-0.25 , .29** | -0.00 , .50 | **-0.25 , .39** |

Table 5: Final weights learned across all model x task combinations, showing [mean, standard deviation] across all recurrent layers. Distributions which are significantly non-zero at the p<.001 level are indicated by bolded values.

## C    APPENDIX: LEARNED WEIGHTS AND AUTAPSES

| Task | Heaviside | Sign | Tri-Value |
|---|---|---|---|
| GSC | 80.49 | 79.13 | 82.75 |

