# OpenReview forum: "Stateful Dynamics for Training of Binary Activation Recurrent Networks"
_ICLR.cc/2025/Conference — Submitted to ICLR 2025_

### Official Review · Reviewer_cdT5 · 2024-10-29

**Soundness:** 1
**Presentation:** 2
**Contribution:** 1
**Rating:** 3
**Confidence:** 4

**Summary:**

This paper deals with artificial neural networks with binary activations (BANNs). Spiking neural networks (SNNs) are a subset of BANNs. The authors investigate whether methods to train feedforward BANNs (e.g., surrogate gradient)  also work in a particular kind of recurrent BANNs: SNN with recurrent connectivity, with and without (leaky) integration. It turns out that these methods fail without (leaky) integration.

**Strengths:**

These results are new.

**Weaknesses:**

The scope of the paper is very narrow. Essentially, the authors take the sort of architectures that is typically used by the SNN community and show that the usual training methods (e.g., surrogate gradient) fail when removing the (leaky) integration (this is somewhat useful to know for the SNN community, but the vast majority of papers use integration anyway because it is useful to learn temporal dependencies). However, recurrent BANNs are a much broader class. For example, binarized GRU has been proposed (see SpikGRU by Dampfhoffer et al), as well as binarized LSTM (https://ieeexplore.ieee.org/abstract/document/7743581). So much more work would be needed to support their general claim that integration is necessary and sufficient to train recurrent BANNs.

Minor points:

* The SNN community always uses {0,1} activations, but the BANN community use {-1,1} most of the time. This should be discussed. In the experiments, the author restricts themselves to {0,1} activations. This again restricts the scope.

* The accuracy they reach is well below the SOTA (e.g., around 80% for GSC vs 95% here https://openreview.net/forum?id=4r2ybzJnmN)

* "g_L is a term which regulates the speed with which x_L decays to zero in the absence of inputs"
tau_x already does that. One constant is enough.

* Eq 9: I think it should be dx_L / dt

* You may want to say that Eq 12 corresponds to the (non-leaky) Integrate and Fire (IF) neuron.

* Eq 13 bottom: I think it should be dx_L, not dx_L / dt

**Questions:**

>For networks with temporal dynamics, the entire image was presented for 16 timesteps and the network output was taken as activity on the final step.

It's more common to take the mean or max activity across timesteps. Have you tried?

---

> ### Author Response · Authors · 2024-11-28
>
> We thank the reviewer for their useful comments, particularly on additional literature sources which are now cited. We have uploaded a revised manuscript which has a significantly expanded introduction section. While the scope of the paper has not changed from the original submission, we are hopeful that the revised introduction provides a better context for our work, while also providing a justification for the investigation of the – implicitly understood but seemingly not explicitly investigated – essential differences between SNNs and the broader class of BARNNs.
>
> > The scope of the paper is very narrow. Essentially, the authors take the sort of architectures that is typically used by the SNN community and show that the usual training methods (e.g., surrogate gradient) fail when removing the (leaky) integration (this is somewhat useful to know for the SNN community, but the vast majority of papers use integration anyway because it is useful to learn temporal dependencies).
>
> We have significantly expanded the introduction section to discuss the differences in “explicit” (via recurrent weights) and “intrinsic” (via temporal dynamics of units) recurrence. While intuitively the integrative state of SNNs learns short-term temporal dependencies, the significantly revised section 4.1 highlights that integrative state is essential for training of the explicit recurrence as well.
>
> > However, recurrent BANNs are a much broader class. For example, binarized GRU has been proposed (see SpikGRU by Dampfhoffer et al), as well as binarized LSTM (https://ieeexplore.ieee.org/abstract/document/7743581). So much more work would be needed to support their general claim that integration is necessary and sufficient to train recurrent BANNs.
>
> We appreciate the reviewer’s reference to the literature and have added these relevant citations to the work. We note that the works support the claims of our paper:
>
> The SpikGRU has similar properties to the work presented here, in that the pre-activation states (`i’ and `v’ in their case) are real-valued integrations, while only the binary `s’ is transmitted. We have now cited this as related work.
>
> Edel & Koppe does not appear to binarize the activity or weights of the recurrent layers (“Algorithm 1”, line 5 of the cited work), but instead binarize only the feedforward layers and keep real-valued LSTM layers.
>
> We believe this emphasizes the need for the explicit discussions of our paper. The literature seems to omit or work around the issues in binary activation recurrent layers, while ours makes explicit what seems to be an implicit knowledge of the community (that the pre-binarization state  must be real-valued).
>
> > The SNN community always uses {0,1} activations, but the BANN community use {-1,1} most of the time. This should be discussed. In the experiments, the author restricts themselves to {0,1} activations. This again restricts the scope.
>
> We appreciate the comment on how the chosen activation values may affect the generality of the results. We have run added a small experiment on [{0,1}, {-1,1} and {-1,0,1}] values, all using the STE surrogate on the GSC task. We found not systematic differences in performance suggesting that, as with the reset and leak dynamics, the integrative state mechanism allows training that is robust to chosen activation values.
>
> > The accuracy they reach is well below the SOTA (e.g., around 80% for GSC vs 95% here https://openreview.net/forum?id=4r2ybzJnmN)
>
> We have added a citation to the suggested work, as well as others which have used SNNs on the GSC and CIFAR10 datasets. We do note however that those papers use additional mechanism such as trained delays. Our work investigates the fundamental structure of recurrence and implications for training, but should be able to be combined with additional mechanisms.
>
> > "g_L is a term which regulates the speed with which x_L decays to zero in the absence of inputs" tau_x already does that. One constant is enough.
>
> We have highlighted that g_L allows leaks to be turned on (1) or off (0), switching the neuron between a leaky integrator and a pure integrator. This is an important distinction to tau which could not turn this dynamic on/off. In a more general sense, changing g_L allows the leak rate to be tuned semi-independent of the accumulation rate.
>
> > You may want to say that Eq 12 corresponds to the (non-leaky) Integrate and Fire (IF) neuron.
>
> We have added a brief explanation of the relationship of this equation the LIF, as well as the functionality of these dynamics.
>
> Again, we are thankful for the important points that the reviewer had contributed, and which led to the significantly expanded introduction which we are hopeful provides a better justification for this work in the broader context of binary activation networks, rather than a more narrow scope of only considering SNN properties.

---

> > ### Comment · Reviewer_cdT5 · 2024-11-28
> > **How about my question?**
> >
> > I thank the authors for their rebuttal.
> > However, it seems that they did not answer my question.

---

> > > ### Author Response · Authors · 2024-11-28
> > > **Re: How about my question?**
> > >
> > > The choice of readout mechanism, including whether taking the maximum or last value of the read-out units, is a hyperparameter. The SNN community appears to use both mechanisms interchangeably (eg see https://ieeexplore.ieee.org/abstract/document/10242251 for an overview of such choices). The choice of last-step decoding is also more similar to the step-by-step decoding required for the tracking task which has a target value on each timestep.
> > >
> > > Because neither maximum nor final step decoding is universally used, and the trained networks are successfully trained using the last-step approach, we do not believe that the choice to only investigate final-step decoding constitutes a limitation of the study.

---

> > > > ### Comment · Reviewer_cdT5 · 2024-11-28
> > > > **Final rating**
> > > >
> > > > > the trained networks are successfully trained using the last-step approach
> > > >
> > > > Well, "successfully trained" is a subjective statement. As I said in my first review, the accuracy is well below the SOTA. I think if the authors could do their analysis on SOTA networks, it would make the paper more interesting.
> > > >
> > > > Anyway, this is not my main criticism.
> > > > As I said in my first review, my main criticism is the narrow scope of the paper.
> > > > I thank the authors for their rebuttal, and I think the manuscript has been improved, so I'm raising my score to 3.
> > > > However, in my opinion, this paper remains below the acceptance threshold due to its narrow scope.

---

### Official Review · Reviewer_TLaJ · 2024-11-04

**Soundness:** 3
**Presentation:** 3
**Contribution:** 3
**Rating:** 8
**Confidence:** 3

**Summary:**

Efficient recurrent processing is increasingly important for energy or memory-sensitive spatiotemporal processing tasks. RNNs with binarized activations (BARNNs) would provide increased efficiency. However, training binary recurrent RNNs is generally regarded as difficult in the existing literature.

The authors illustrate on a keyword spotting task that conventional BARNNs have non-smooth temporal gradients, while a floating point RNN and recurrent LIF spiking neural network (SNN) have smoother temporal gradients. The authors hypothesize that these smoother gradients are beneficial to learning.

The authors reproduce the difficulty of training BARNNs. They apply three existing methods: (1) surrogate gradients (STE), (2) probabilistic activations, and (3) sharpening activations over training. Importantly, the authors show that on a static-input task, CIFAR, BARNNs train comparably well compared to a floating-point baseline. In contrast, for spatiotemporal tasks SC and SOT, BARNNs do not train well compared to a floating-point baseline. Interestingly, however, in contrast to the 3 conventional BARNN methods listed above, the authors train a LIF SNN and achieve competitive task accuracy on all three tasks compared to a floating-point baseline. The authors identify the stateful accumulation, leaky, and reset mechanisms as potential explanators for the SNN’s advantage over the conventional BARNN methods.

The authors hypothesize that the stateful accumulation is responsible for the SNN advantage, so they add stateful accumulation to the pre-activations of the 3 conventional BARNN methods and recover competitive task accuracy with the LIF SNN and floating point baseline for SC and SOT tasks, for all but the sharpening method. This evidence supports the hypothesis regarding the critical role of stateful accumulation.

The authors also investigate how the leak and reset features of LIF SNNs affect SSNs trained using surrogate gradients. The authors train networks using surrogate gradients or probabilistic activations with stateful accumulation (integration), leak, and/or reset. The authors find that generally competitive task performance is maintained in all cases, and they conclude that the key ingredient for well-performing BARNNs is stateful accumulation (integration). Furthermore, the authors find that the distributions of trained parameters varies among the different configurations.

**Strengths:**

Significance.
This work makes a valuable connection between conventional binarized networks and spiking neural networks (SNNs). The connection is particularly valuable because it carefully uncovers “all you need” to get the benefit from SNNs in more conventional binarization approaches for recurrent networks – namely, stateful accumulation in preactivations (integration).

Originality.
This work is the first I have seen that systematically compares conventional BARNN training methods for RNNs to SNNs on relevant spatiotemporal tasks.

Quality.
The author’s approach is generally clear, and their line of reasoning generally lucid.

Clarity.
The state goal and subsequent structure of the paper creates a clear narrative illustrating how the author’s reached their findings.

**Weaknesses:**

I noted the following weaknesses:

Notably, the SNN Eq (7) has an infinite-extent surrogate derivative, while Eqs (2) (3) and (4) for BARNNs have finite-extent surrogate derivatives. One confounding reason for why the SNN performs better on spatiotemporal tasks, in addition to the integrative state, is the infinite-extent surrogate derivative. Could this also be the reason why SNNs learn better than the conventional BARNN approaches? Or stated another why – why did the authors choose finite-extent surrogate derivatives for BARNNs and infinite-extent for SNN? Stated yet another way – is there a reason why this finite-vs-infinite extent distinction is irrelevant?

In section 4.1, the authors state that BARNNs are unstable through time. In what sense are they unstable - are the authors using ‘stability’ in some a technical sense? E.g., one could argue that the dynamics are in fact stable – they do not go to infinity nor negative infinity.

I have trouble following the logic from line 313 to 341. For instance, why would activities propagate poorly in BPTT in the oscillatory example Eq 10?  The surrogate derivates are not zero, so as far as I can tell, gradients would propagate without issue. In line 325, why would taking the surrogate gradient of this patter with respect to the recurrent weights provide minimal information other than the relative value of the recurrent weights to the feedforward activity? In line 327, what’s a “real valued” BARNN? My understanding was that BARNNs had binary activations by definition. In line 338, “resulting in dense discontinuities in the input” – to what input do the authors refer? More generally regarding the choice of a single-neuron BARNN illustrative example – are there no averaging effects when many neurons are considered that could help smooth out binary activation oscillations?

**Questions:**

I asked the most salient questions above in the “Weaknesses” section. The questions that follow are more minor.

1.	To be clear, are all weights and integrative states floating point in this work? (Only activations are binary.)

2.	Why are the dense layers for SC and SOT not recurrent?

3.	Is there anything that can be said about the hyperparameter selection process used in this work, to help justify that the conclusions drawn in this work are not an artifact of certain hyperparameter choices? (E.g., perhaps the reasoning sharpening did not work as well as other BARNN methods is because it requires different hyperparameter settings to perform well.)

4.	What is an autapse?

5.	Regarding line 402, the authors state distributions of leaks is beneficial. Did the authors use a distribution of leaks in this work? Were leaks trainable parameters?

Thank you for this fascinating work.

---

> ### Author Response · Authors · 2024-11-28
>
> We thank the reviewer for their complete summary and general regard for the impact of the work. We appreciate the specifically noted weaknesses and have incorporated changes into the text as appropriate, with itemized responses below:
>
> > Notably, the SNN Eq (7) has an infinite-extent surrogate derivative, ...
>
> The SNN community has utilized several surrogate functions, including a mixture of infinite-extent and limited-extent methods, and there does not seem to be a systematic difference in performance (see Neftci, Mostafa, Zenke 2019). We do note that one evaluated model (Table 4, Surrogate 1 1 1 – containing leaks, state, and explicit reset) is essentially an SNN trained with the finite-range surrogate gradient, and does not show lower performance than the LIF model.
>
> > In section 4.1, the authors state that BARNNs are unstable through time. In what sense are they unstable - are the authors using ‘stability’ in some a technical sense? E.g., one could argue that the dynamics are in fact stable – they do not go to infinity nor negative infinity.
>
> This was a regretful choice of wording on our part. We have changed the terminology from “stable” to “smooth”. The intent of section 4.1 was to demonstrate temporal smoothing of activity, not stability in the technical sense.
>
> > I have trouble following the logic from line 313 to 341. For instance, why would activities propagate poorly in BPTT in the oscillatory example Eq 10? The surrogate derivates are not zero, so as far as I can tell, gradients would propagate without issue. In line 325, why would taking the surrogate gradient of this patter with respect to the recurrent weights provide minimal information other than the relative value of the recurrent weights to the feedforward activity?
>
> We have removed this “argument by demonstration” with a new Equation 12, which examines the mechainsm of BPTT for recurrent weights – highlighting the need for a differentiable temporal derivative of the pre-activation state. By Figure 4.1 then demonstrates that BARNNs do not have this property, explaining the inability to sufficiently train them.
>
> > In line 327, what’s a “real valued” BARNN? My understanding was that BARNNs had binary activations by definition.
>
> We have corrected this to “stateful BARNN”. As with the sections following, the state is real-valued (thus the previous mistake)
>
> > To be clear, are all weights and integrative states floating point in this work? (Only activations are binary.)
>
> Yes, this work only addresses binary activations. We have added a sentence to the conclusions regarding binary weights.
>
> > Why are the dense layers for SC and SOT not recurrent?
>
> This choice was based on previously published architectures which were able to perform these tasks in SNN approaches. At a conceptual level, having only the early layers be recurrent demonstrates that sufficient temporal information is extracted by a single recurrent layer and simply needs to be read out by feedforward transformations. We have added a citation for these
>
> > Is there anything that can be said about the hyperparameter selection process used in this work, to help justify that the conclusions drawn in this work are not an artifact of certain hyperparameter choices? (E.g., perhaps the reasoning sharpening did not work as well as other BARNN methods is because it requires different hyperparameter settings to perform well.)
>
> We did not generally perform hyperparameter optimization, and instead used architectures from prior publications, default learning rates, etc. The exception to this is in the sharpening approach, for which we did perform hyperparameter optimization (Appendix A).
>
> > What is an autapse?
>
> We have added an explanation, highlightning that autapses are the diagonal elements of recurrent weight matrices.
>
> > Regarding line 402, the authors state distributions of leaks is beneficial. Did the authors use a distribution of leaks in this work? Were leaks trainable parameters?
>
> We utilized only untrained uniform leaks in the current work. We have revised this sentence slightly to emphasize that we are only justifying why such parameters might be useful in general.

---

### Official Review · Reviewer_SQFd · 2024-11-04

**Soundness:** 2
**Presentation:** 1
**Contribution:** 3
**Rating:** 3
**Confidence:** 2

**Summary:**

The paper investigates the impact of recurrence as an inductive bias in binarized neural networks, revealing that recurrence leads to temporal instability when using modern surrogate gradient methods, in contrast to spiking neural networks. Furthermore, it demonstrates that integrating local dynamic states, similar to those in spiking neural networks, enhances temporal stability in recurrent binarized neural networks.

**Strengths:**

1. **Innovative Application of Spiking Neural Network (SNN) Concepts**
   - Introducing elements from SNNs, like pre-activation state, leakage, and reset mechanisms, is an innovative approach to handling binary activations in RNNs.

2. **Exploration of Multiple Training Methods**
   - The paper systematically compares several training strategies (surrogate gradient descent, probabilistic surrogates, and progressive sharpening), showing a thoughtful approach to exploring solutions for BARNNs.

3. **Comprehensive Experimental Setup**
   - The use of three distinct tasks—image classification, keyword spotting, and small object tracking—demonstrates the versatility of the proposed methods across different types of temporal and spatial data.

4. **Potential for Hardware Implementations**
   -  This is valuable in the context of real-world deployment, where binary networks and reduced precision can offer efficiency gains, particularly for embedded or neuromorphic systems.


With these strengths, the paper lays a foundation for further exploration and potential practical applications in energy-efficient temporal modeling.

**Weaknesses:**

- **Equation Nomenclature and Legibility**:
   - Equations in the paper are difficult to follow due to inconsistent or unclear notation. Key variables are not defined consistently, and some choices create ambiguity. For instance, in Equation 8, it’s unclear if the layer is intended to be interpreted as a stacked ConvRNN. Additionally, the same variable, ‘y’, is used across both the spiking neural network (SNN) and binarized ConvRNN contexts, which conflates distinct mechanisms and makes tracking the model dynamics challenging.

- **Unprincipled Approach in Section 4.1**:
   - The demonstration of binarized recurrent network instability in Section 4.1 lacks theoretical grounding. Beyond the empirical results, the chosen edge case of a constant input does not convincingly justify the instability of these networks. Additionally, Figure 4.1 requires more explanation: it seems to show that the binary activation seems to reconstruct the input, unlike the SNN, could you provide further clarity on this. I am willing to adjust my score if further clarity on this figure is provided.

- **Lack of Focus**:
   - The contributions are listed but lack clarity, and the paper attempts to address multiple aspects of binary recurrent network training without a clear focus. For example, the incorporation of pre-activation states, leak, and reset mechanisms are all discussed but without a strong, unified narrative explaining why each is necessary. This could be solved by strengthen the message and the structure of the paper. The paper could greatly benefit from clarity.

In summary, while the paper addresses a relevant problem, its clarity, and structure could be significantly improved. Addressing these weaknesses would enhance the impact and accessibility of the work.

**Questions:**

See weaknesses which highlight some key questions. In particular, I'd like clarity on Figure 4.1, and Equation 8 in the manuscript.

---

> ### Author Response · Authors · 2024-11-28
>
> We appreciate the reviewer’s receptiveness to the overall concepts demonstrated in the paper, and take note of their comments regarding presentation. We have made substantial changes in the resubmitted manuscript, particularly in the introduction and refocusing section 4.1. We believe that these revisions have strengthened the focus of the paper and would like to thank the reviewer for their helpful comments.
>
> Responses to specific points below point to revisions in the manuscript.
>
> > Equations in the paper are difficult to follow due to inconsistent or unclear notation. Key variables are not defined consistently, and some choices create ambiguity. For instance, in Equation 8, it’s unclear if the layer is intended to be interpreted as a stacked ConvRNN. Additionally, the same variable, ‘y’, is used across both the spiking neural network (SNN) and binarized ConvRNN contexts, which conflates distinct mechanisms and makes tracking the model dynamics challenging.
>
> We appreciate that using the same variable for differing functions overloads them. However, after deliberation we have decided to keep the consistent variable names, with the addition of subscripts to differentiate when necessary (eg: \Theta_{ste}). We have also added in the introduciton a “unified equation” (equation 3) which we believe illustrates the importance of consistently using the nomenclature of table 1. We hope that this added context helps to make it clear that the specific instance of each variable/function differs slightly.
>
> > Unprincipled Approach in Section 4.1: The demonstration of binarized recurrent network instability in Section 4.1 lacks theoretical grounding. Beyond the empirical results, the chosen edge case of a constant input does not convincingly justify the instability of these networks.
>
> We have significantly revised section 4.1 to replace the edge-case with a more general commentary on the evolution of temporal derivatives in recurrent settings. The new equation 12 examines the derivative of recurrent weights with respect to the pre-activation state to demonstrate how a smooth temporal change in this variable is essential for chaining of gradients through time. We have also rephrased this section from “stability” to the more accurate terms “smoothness” “continuous” “discontinuous” etc.
>
> > Additionally, Figure 4.1 requires more explanation: it seems to show that the binary activation seems to reconstruct the input, unlike the SNN, could you provide further clarity on this. I am willing to adjust my score if further clarity on this figure is provided.
>
> In line with the previous comment, we have revised the caption of Figure 4.1 to act as a specific example of temporal (un)smoothness. This then links back to equation 12 to demonstrate that the unsmooth temporal patterns of the BARNN can not sufficiently train.
>
> Additionally, it is important to note that reconstruction of the inputs is not function of the layers illustrated in this figure. Instead, each time slice should be extracting some information on the relationship of input rows.
>
> > Lack of focus: The contributions are listed but lack clarity, and the paper attempts to address multiple aspects of binary recurrent network training without a clear focus. For example, the incorporation of pre-activation states, leak, and reset mechanisms are all discussed but without a strong, unified narrative explaining why each is necessary. This could be solved by strengthen the message and the structure of the paper. The paper could greatly benefit from clarity.
>
> We appreciate the insight on how the previous manuscript did not explain why these various terms were included. We have significantly revised the introduction to discuss how these terms are all present in standard SNNs and that the goal of the paper then is to investigate which of these are essential.
>
>
> Again, we would like to thank the reviewer for their helpful comments, which lead to the significant revision of the introduction and section 4.1. We are hopeful that the revised version more coherently demonstrates the aims of the paper.

---

### Official Review · Reviewer_Rr5v · 2024-11-04

**Soundness:** 2
**Presentation:** 2
**Contribution:** 1
**Rating:** 3
**Confidence:** 4

**Summary:**

This manuscript discusses experiments with various quantization strategies to binarize activations of neural models, particularly recurrent. The authors have included in the list of strategies SNN training by seeing the LIF neuron model as yet another binary yet stateful activation function, and conclude that it is a very effective approach for quantization of recurrent networks but they assess that decay, and reset/refractoriness do not really play any influential role.

**Strengths:**

I could not identify any, i am sorry.

**Weaknesses:**

I find that this manuscript lacks basic understanding of SNNs, does not have a clear scope, and the experimentation lacks depth and structure. Instead, in many parts the authors just re-discover basic concepts or properties about SNNs and recurrent networks.

First and foremost the authors claim that "Binary activation NNs have only been reported for feedforward topologies" (!), when practically every SNN network is a binary activation recurrent network.

The authors also claim as a contribution that state allows to train binary activation recurrent networks, but well isn't that obvious, the state is responsible for the recurrent behavior to begin with ?

What the authors claim to be temporal instability treatise for recurrent layers (in section 4.1), is really just a discussion about the smoothness of the gradient, or am i missing something?

What the authors call different training methods are really one method, only THE backprop (BP) method, and instead they look at different strategies for quantizing (binarizing) activations using BP in-training. In these strategies they test various combinations of statefulness/statelessness, approximations of firing functions (heavyside, noisy heavyside, and hard sigmoid converted to heavyside progressively), and surrogates of the gradients of the binary firing function (actually just one the STE with different gains). However, the combinations are not exhaustively examined but rather haphazardly chosen.

Although the authors claim contributions relevant to recurrent networks, the experiments carried out are not with temporal tasks but rather all spatial. They are also executed in a way (the inputs are not provided sequentially but in a single timestep) that the authors only observe the step response of the models (as dynamical systems) and not the temporal integration of the data dynamics, which makes no sense to me.

Moreover the results they present in two tables hardly support their claimed contributions, in different datasets different strategies give the best results, and it is by no means decisive that statefulness attains the best result (but then again also the tests are not temporal either).

Finally, exactly because the choice and design of experiments (with non temporally integrated stimulus) I would not expect to see any effect from decay or refractoriness, so I wonder what makes the authors conclude that these play not role whatsoever in general ?

Additionally

In l-099 the authors try to justify they choice for centering the activation functions, without explaining why is that relevant.

In l-100 the authors talk about literature standards without explaining what standards they refer to.

In Table 2 the difference between CNN-RNN and CRNN has not been explained

The SOT benchmark is not explained clearly

**Questions:**

See the Weaknesses section.

---

> ### Author Response · Authors · 2024-11-28
>
> We regret that many of the points of the paper were not clear in the previous version of the manuscript. We have uploaded a significantly revised version of the manuscript that addresses many of the points raised by the reviewer. We believe that the revisions, particularly the significantly expanded introduction section, should highlight the relationship between SNNs and the broader class of binary recurrent ANNs. We have responded to specific comments below, either highlighting changes made to the manuscript or drawing attention to where these points were previously made.
>
> > I find that this manuscript lacks basic understanding of SNNs, does not have a clear scope, and the experimentation lacks depth and structure. Instead, in many parts the authors just re-discover basic concepts or properties about SNNs and recurrent networks.
>
> We regret that the intended contributions of the work were not clear. We have rephrased the introduction to emphasize that the work is intended to be a systematic deconstruction of SNNs, to illustrate what properties of the SNN are essential for training in recurrent topologies.
>
> > First and foremost the authors claim that "Binary activation NNs have only been reported for feedforward topologies" (!), when practically every SNN network is a binary activation recurrent network. The authors also claim as a contribution that state allows to train binary activation recurrent networks, but well isn't that obvious, the state is responsible for the recurrent behavior to begin with?
>
> These two comments highlight how both explicit recurrent connections within a layer, and intrinsic dynamics of individual units are both referred to as “recurrent networks”. For instance, an SNN with purely feedforward weights may still be referred to as a recurrent network, due to the temporal dependence of the internal state. We have added a paragraph to the introduction clarifying these as two separate properties. We have also emphasized that our overall finding is in fact that the dynamics are what enable explicit recurrent connections under the constraint of binary activation.
>
> > What the authors claim to be temporal instability treatise for recurrent layers (in section 4.1), is really just a discussion about the smoothness of the gradient, or am i missing something?
>
> We have rephrased section 4.1 to refer to “temporal smoothness” instead of “temporal stability”. We have additionally reworked this example to refer more directly to the temporal derivatives, rather than working by example. By working with the partial derivative for recurrent weights w.r.t. recurrent state we highlight how a smooth state is essential for chaining gradients through time.
>
> > What the authors call different training methods are really one method, only THE backprop (BP) method, and instead they look at different strategies for quantizing (binarizing) activations using BP in-training. In these strategies they test various combinations of statefulness/statelessness, approximations of firing functions (heavyside, noisy heavyside, and hard sigmoid converted to heavyside progressively), and surrogates of the gradients of the binary firing function (actually just one the STE with different gains). However, the combinations are not exhaustively examined but rather haphazardly chosen.
>
> We have added an emphasis to “training BANNs” to point out various non-backprop based methods that exist. However, we also add a point that we choose to focus on BP-variant methods, as they are more common in the ML literature.
>
> > Although the authors claim contributions relevant to recurrent networks, the experiments carried out are not with temporal tasks but rather all spatial. They are also executed in a way (the inputs are not provided sequentially but in a single timestep) ... (with non temporally integrated stimulus) I would not expect to see any effect from decay or refractoriness, so I wonder what makes the authors conclude that these play not role whatsoever in general ?
>
> We are sorry that the reviewer misinterpreted the tasks as spatial and non-temporal. While the CIFAR10 task is not temporal, the other two tasks (audio classification and video tracking) are. In the CIFAR10 task for the temporal networks the stimulus is presented for multiple timesteps. We now explicitly state that the speech command dataset is a “temporal classification task” and that the small object task is a “video”.
>
> >In Table 2 the difference between CNN-RNN and CRNN has not been explained
>
> We now provide a link to equation 10, which outlines the CRNN structure.

---

> > ### Comment · Reviewer_Rr5v · 2024-11-28
> >
> > I would like to thank the authors for putting an effort to revise the paper.
> >
> > The introductory parts and some of the claims have been clearly improved and corrected.
> > I have not read the entire new manuscript version yet, but peeking in it I still see some important flaws, the most important of which is the out of scope experimentation!! (as explained more below)
> >
> > > We are sorry that the reviewer misinterpreted the tasks as spatial and non-temporal. While the CIFAR10 task is not temporal, the other two tasks (audio classification and video tracking) are. In the CIFAR10 task for the temporal networks the stimulus is presented for multiple timesteps. We now explicitly state that the speech command dataset is a “temporal classification task” and that the small object task is a “video”.
> >
> > The task in the dataset may be temporal in its nature when one of the dimension is time (e.g. SC or SOT). But the way you provide it to the network is not temporal at all! Lets break it down to the basics:  You have an adaptive system with state (call it dynamical system, or feedback control system, or just SNN, it is all the same), right?. The impulse response of that system tell you something about its behavior (of the system not of the stimulus) when you stimulate it for only one timestep and let it reverberate for N timesteps thereafter. The step response of that system tells you something about its behavior (again of the system not of the stimulus) when you stimulate it and keep it stimulated with the same stimulus as it reverberates for N timesteps thereafter. Does that sound familiar with what you do in your experiments ?  HOWEVER, if you want to see how the system processes a temporal stimulus, you need to provide as input a time-varying (across timesteps) signal! Then you see the temporal interaction of your system with the stimulus. In other words in the case of the SC par example, you need to provide a different column of a spectrogram at each timestep, in order to benefit from the temporal attractor dynamics of your RNN/SNN. At the end of the day this makes your network more compact (smaller input dim) which is why you should/would use an RNN/SNN in the first place. Put it in another way if you provide the whole spectrogram in one timestep to model, the model does not understand the semantics of each axis anyway, from its point of view it is all one image, and this makes the task spatial !. Same goes for the video, if you dont provide a different frame as input in each timestep.
> >
> > Conclusion from this is that you are testing an RNN/SNN in 3 datasets (2 of which are temporal) as spatial tasks. I.e. in 3 spatial tasks. This I believe biases significantly your conclusions and observations.
> >
> > > Contribution 1: Illustrate temporal discontinuities for binary activation explicit recurrent layers, leading to
> > unsuccessful backpropagation through time (section 4.1).
> >
> > This is definitely no news. That is why the approximations of the spiking operation were introduced in the first place as well as the surrogate gradient. Look up the literature on spatio-temporal back-propagation (https://arxiv.org/abs/1706.02609), surrogate-gradient training (https://arxiv.org/abs/1901.09948) and related literature and analysis therein.
> >
> > > Contribution 2: Demonstrate that surrogate gradient methods fail to converge when employed with a binary
> > activation in a recurrently connected layer (section 4.2)
> >
> > I fail to see how you conclude this. All you observed in this section is that a specific choice of surrogate gradient function does not reach the score you hoped in 2-3 specific image processing tasks (explained why in the prev comment). But absence of convergence ?  ... and generalizable to all possible surrogates ? Moreover the refence in you main text from Bengio's team showed that the STE does work even in absence of recurrent state, no? (and I can also attest from personal experience that I never had convergence problems from using STE as a surrogate).
> >
> > > Contribution 3: Demonstrate, across multiple surrogate approaches, that incorporating pre-activation integrative state allows training of recurrent binary activation networks (section 4.2).
> >
> > Well but if you don't have recurrent state (what you call pre-activation integrative state) you don't have recurrent networks to begin with, or am i missing something? Are you referring to the explicit recurrent connections as what characterizes recurrent nets even if there is not somatic state ? If so did you experiment with that and how ? (because a recurrency is equivalent to have memory of 1 timestep back).
> >
> > And what multiple surrogate approaches are you referring to. You only present one surrogate function.
> >
> > > Contribution 4: Show robustness of performance when including additional state dynamics such as explicit
> > reset and proportional leakage of sub-threshold state (section 4.3)
> >
> > These are no new contributions, and they are not even universal. They depend on combination of task and parameterisation.

---

> ### Author Response · Authors · 2024-11-28
>
> We would like to thank the reviewer for continuing to engage with the revised paper. Below are responses to each of the new points, but I believe the first is a very important one, and that understanding the way our stimuli were presented and the networks evolve will alleviate many later concerns.
>
> >The task in the dataset may be temporal in its nature when one of the dimension is time ... you need to provide as input a time-varying (across timesteps) signal!
>
> This is exactly how the stimuli are presented to the network. Per line 320 - 321 "on each timestep all 64 frequency bands **a single column of Figure 3B frames**". Similarly with the SOT: "readout location averaged across a given 100-frame trial". That is, the networks evolve with temporally evolving inputs and outputs -- 64 timesteps in the case of the SC and 100 timesteps in the case of the SOT. We do maintain spatial structure in these inputs by the (spatial) convolutional weights, but that makes these spatiotemporal networks, not spatial.
>
> While the CIFAR10 example presents the same stimulus on multiple timesteps and may not be a "temporal task" in that regards, it is important to note that this task is used as a **contrast** to the temporal tasks. As noted in the discussion of Table 2, the baseline BARNN networks perform on-par with the FP networks for this task, precisely because there is no need to extract temporal information, whereas they fail in the GSC and SOT tasks which do require temporal processing.
>
> >Contribution 1 ...This is definitely no news.
>
> The works you provided utilize smooth temporal state by the membrane state, while no previous studies have shown temporal BARNN training without the state. What we have done in the current work is to take all of the differences between the simplest LIF-based models and a pure BARNN and remove them one by one. We believe that **explicitly** showing that training without state is an important contribution for the development of BARNN in the future. By drawing attention to  parallels between quantized (explicit) recurrent networks and SNNs, we hope to allow cross-talk between groups that appear to be operating in parallel with each other.
>
> > Contribution 2:
> We have not found a case were STE on binary activation was performed in recurrent (in this case "explicit" recurrence through weight matrices) has been performed, and the current work (table 2, bottom section) confirms this. While we only utilized three training approaches, we also provided an explanation for why any surrogate activation function, which by definition can not smooth the **temporal** discontinuities of BARNNs will fail (section 4.1).
>
> > Contribution 3:
>
> Yes, a set of feedforward units that are connected by a recurrent weight are considered "explicitly recurrent". Elucidating the differences between the explicit recurrence used in ANNs and the intrinsic recurrence of SNNs (state / membrane voltage / etc) is the purpose of the new "explicit versus intrinsic recurrence" section of the introduction.
>
> > multiple surrogate approaches are you referring
>
> We utilized both the STE and probabilistic approaches, which may be thought of as multiple surrogates. We would be open to clarifying this text before a camera-ready version of the paper.

---

> > ### Comment · Reviewer_Rr5v · 2024-11-29
> >
> > > This is exactly how the stimuli are presented to the network. Per line 320 - 321 "on each timestep all 64 frequency bands a single column of Figure 3B frames".
> >
> > I see, then fine. To me this is not a clearly written though. How about .. "on each timestep all 64 frequency bands of a single column of a frame shown in Figure 3B".
> >
> > > The works you provided utilize smooth temporal state by the membrane state, while no previous studies have shown temporal BARNN training without the state. What we have done in the current work is to take all of the differences between the simplest LIF-based models and a pure BARNN and remove them one by one. We believe that explicitly showing that training without state is an important contribution for the development of BARNN in the future. By drawing attention to parallels between quantized (explicit) recurrent networks and SNNs, we hope to allow cross-talk between groups that appear to be operating in parallel with each other.
> >
> > You keep on referring to training a BARNN without taking account of the state, but if you remove the state you don't have an RNN to begin with. Even if you have very fast decay of the LIF state, the explicit recurrency will still reinforce some information about the previous timestep (and if your recurrent weight equals the finiring threshold then you maintain all previous state). And the fact that you have not seen it in literature, maybe because it is too obvious ? Had your results shown something different then I would say maybe you have contribution.
> >
> > > We have not found a case were STE on binary activation was performed in recurrent (in this case "explicit" recurrence through weight matrices) has been performed, and the current work (table 2, bottom section) confirms this. While we only utilized three training approaches, we also provided an explanation for why any surrogate activation function, which by definition can not smooth the temporal discontinuities of BARNNs will fail (section 4.1).
> >
> > Pragmatically and formally you can see the STE as the derivative of the heavyside function (subject to constraints), and in this case it has nothing to do with state. You can apply it only when there are spikes (i.e. use spikes as gating for the error grad), or you can ignore the spikes (since they are not differentiable) and use it as the (surrogate) derivative of the loss over the membrane state. The difference lies on what assumption you make for spikes, a heavyside or a dirac delta. Regarding the membrane state you can decide whether your decay time constant is such that state is forgotten immediatelly and you are in BNN turf or whether your time constant allows you to keep track of the past and you are in BRNN turf. Finally recurrent state can be reinforced locally only (either explicitly -- 1 timestep back -- and/or implicitly many timesteps back) in which case your weight matrix is diagonal, or laterally in which case your weight matrix is non-zero off-diagonally. This type of thinking unifies formally BNNs BRNNs and SNNs. So I m not sure I understand what you are trying to claim as unique or new, the fact that you looked at a special regime of parameters in this ? That would be fine if you had made a surprising/unexpected discovery in that special case, but with all respect to your work, I don't much else than confirming something expected.
> >
> > > Yes, a set of feedforward units that are connected by a recurrent weight are considered "explicitly recurrent". Elucidating the differences between the explicit recurrence used in ANNs and the intrinsic recurrence of SNNs (state / membrane voltage / etc) is the purpose of the new "explicit versus intrinsic recurrence" section of the introduction.
> >
> > Fine, but is this your contribution ? I remember the first time I heard about it was by Sejnowski in what would be textbook knowledge for SNNs (last time I read it was in https://arxiv.org/abs/1901.09948 and it indeed created an aha effect for many because of contextualization with the surrogate gradient). An RNN in ANNs has also intrinsic recurrence. That is the role of the hidden state matrix.
> >
> > > We utilized both the STE and probabilistic approaches, which may be thought of as multiple surrogates. We would be open to clarifying this text before a camera-ready version of the paper.
> >
> > So you re claiming that this is sufficient to draw generalizable conclusions for all possible surrogates ?

---

> > > ### Author Response · Authors · 2024-12-04
> > >
> > > > I see, then fine. To me this is not a clearly written though. How about .. "on each timestep all 64 frequency bands of a single column of a frame shown in Figure 3B".
> > >
> > >
> > >
> > > We will incorporate this minor change in our local version; however, PDF change submissions are currently locked. This minor change in wording however does not affect the contribution of the work, and certainly does not stem from a “lack of a basic understanding of SNNs”. Given that your initial critique of the paper was based largely on this critical misunderstanding, we ask that you reevaluate the submission in light of this new understanding, as well as the significant revisions submitted last week.
> > >
> > >
> > >
> > > The remainder of your comments demonstrate the criticality of carefully and explicitly drawing the differences between pre-activation and post-activation recurrence, as is now carefully outlined in the introduction and section 4.1. While the findings of our numerical experiments may be the expected outcome by those in the field, it is essential that they be demonstrated in clear mathematical form, lest terms be confused. As an example, the term "state" can refer to the output of a neural layer, or the pre-activation "membrane potential". The previous comment states that the output, through recurrent weights, can constitute ``state`` and therefore support gradients through time. However, the post-activation value can not support smooth gradients through time, as demonstrated in the third partial derivative of the RHS of equation 12 and evaluated empirically in Table 2. This separation of smoothing of activation functions (e.g. STE) and temporal state (`v`) is an important distinction when the pre-activation state has a higher numerical precision than the output of the layer. What we term the "intrinsic recurrence" and "explicit recurrence" is therefore an important distinction, which can not be wrapped in to simply "recurrent". Regarding the referenced work of Neftci et al, our contributions are beyond the scope of that work. While the cited work unifies implicit and explicit recurrence into a single equation  (eq 6 of their work), did not include the temporal recursive dependence of state in the backpropagation through time (their equation in Figure S2), and therefore critically can not point out the issues surrounding temporal continuity of pre-activation state.
> > >
> > >
> > > We have addressed the specific technical concerns raised in the initial round of reviews and added substantial clarity to the paper through the addition of significant introductory material and replaced the original "proof by example" of section 4.1 with the formal evaluation of backpropagation through time (equation 12) in context of surrogate gradients. Regarding scope and impact, we reiterate that formalizing and consolidating ideas are important contributions. Even if our results are intuitive, formalizing such explanations is necessary to draw more general conclusions.

---

### Author Response · Authors · 2024-12-04
**Closing Remarks**

We would like to thank the reviewers for their engagement during the discussion period, which have resulted in a highly revised version of the manuscript which more clearly highlights the primary contribution of the paper: providing a “valuable connection between conventional binarized networks and spiking neural networks” through a “comprehensive experimental setup”,  combined with a theoretical explanation for which of these differences are critical. Having addressed these critical points, particularly by significantly expanding the explanation of intrinsic versus explicit recurrence and a principled investigation of temporal continuity of gradients, has significantly increased the quality of the paper. We believe that the paper now presents a more cohesive and general story and addresses all of the technical concerns of the reviewers.


There remain concerns with the scope and impact of the paper, with an even split between the reviewers viewing the work as either intuitive and narrow, versus general applicable and high impact. We believe that while the overall message of the paper, that integrative state enables smooth gradients with respect to time, may seem intuitive, it is critical to investigate and formalize the underlying reason for such findings. Importantly the previous work that reviewer Rr5v refers to does not investigate the partial derivatives with respect to previous pre-activation state, and therefore does not draw the explicit conclusion that temporal smooth state is necessary. The works cited by reviewer cdT5 meanwhile keep pre-activation state (SpikGRU) or do not binarize the outputs of recurrent layers. This highlights the consistent approach for papers to circumvent issues of temporal differentiability of local state. However, without a formal description of this requirement these approaches may be seen as a "design choice" rather than a fundamental requirement of BPTT. We therefore continue to believe this work is important contribution to the field.

---

### Meta-Review · Area_Chair_vfCL · 2024-12-23

**Metareview:**

This submission provides insights into the role of integrative states in training binary recurrent neural networks (BARNNs) and draws parallels to spiking neural networks (SNNs). Three out of four reviews voted for rejection. Mainly, the scope was found to be narrow. Other issues raised include experimental design issues, such as limited temporal implementation of tasks and below-state-of-the-art performance. Overall, the work requires broader scope, stronger experiments, and competitive results to be admitted to this selective conference.

**Additional Comments On Reviewer Discussion:**

The authors provide a good summary of the discussion in the rebuttal period. The final decision was mostly based on the reviewers' final overall assessments.

---

### Decision · Program_Chairs · 2025-01-22

Reject